# Quasi: a synthetic Question-Answering dataset in Swedish using GPT-3 and zero-shot learning

**Dmytro Kalpakchi**
Division of Speech, Music and Hearing
KTH Royal Institute of Technology
Stockholm, Sweden
dmytroka@kth.se

**Johan Boye**
Division of Speech, Music and Hearing
KTH Royal Institute of Technology
Stockholm, Sweden
jboye@kth.se

## Abstract

This paper describes the creation and evaluation of a synthetic dataset of Swedish multiple-choice questions (MCQs) for reading comprehension using GPT-3. Although GPT-3 is trained mostly on English data, with only 0.11% of Swedish texts in its training material, the model still managed to generate MCQs in Swedish. About 44% of the generated MCQs turned out to be of sufficient quality, i.e. they were grammatically correct and relevant, with exactly one answer alternative being correct and the others being plausible but wrong. We provide a detailed analysis of the errors and shortcomings of the rejected MCQs, as well an analysis of the level of difficulty of the accepted MCQs. In addition to giving insights into GPT-3, the synthetic dataset could be used for training and evaluation of special-purpose MCQ-generating models.

## 1 Introduction

OpenAI's GPT-3 (Brown et al., 2020) is the current state-of-the-art model for text generation. One of the more impressive properties of this model is the way it can perform natural-language tasks without any labeled examples, with so-called *zero-shot learning*. In the case of GPT-3, zero-shot learning entails that the model receives a prompt describing the task verbally (for example, "Translate from English to Spanish"), some input data for the task (a text in English in this example), and then produces the output (the Spanish translation in the example).

Most research using GPT-3 has focused on English, because the bulk of GPT-3's training data (92.6% of words) is English text. Only 0.11%[1]

of the training data is Swedish text. This might sound insignificant at first, but it actually amounts to 220.9 million words, which is quite a sizeable corpus! In addition, Swedish and English are both Germanic languages, so it is possible that some cross-lingual learning has taken place during training. Taking all this into account, we want to test whether GPT-3 would be able to handle tasks in Swedish in a zero-shot fashion. Specifically, the article has the following two goals.

1. Provide a pilot evaluation of GPT-3's ability to generate multiple-choice questions (MCQ) in a zero-shot manner.

2. Create the first synthetic dataset of MCQs, called *Quasi*,[2] for testing reading comprehension of adult language learners of Swedish.

An MCQ consists of a text, a question (called the *stem*) on the text, and a set of answer alternatives, of which exactly one is correct (called the *key*) and all the others are wrong, but plausible (called *distractors*). As we will show, GPT3 is good but far from perfect in generating Swedish MCQs from a Swedish text: more than half of the generated MCQs were incorrect, sometimes in subtle ways. This means that GPT-3 does not provide an ultimate solution to the MCQ-generation task, and that special-purpose models are still required. The synthetic dataset presented here could potentially be used as extra training material for such special-purpose models.

In this paper, we provide a detailed analysis of the errors and shortcomings of the rejected MCQs, as well as an analysis of the level of difficulty of the accepted MCQs, giving insights into the strengths and weaknesses of GPT-3.

---

[1]Hyperlink to a CSV file with the training data statistics

[2]Raw data, annotations, the details on the annotation setup, and the source code are available at https://github.com/dkalpakchi/Quasi

## 2 Related work

The idea of creating synthetic datasets is not new both in NLP in general (Gessler et al., 2020; He et al., 2022), and for Question Answering (QA) specifically (Alberti et al., 2019). To the best of our knowledge, no synthetic datasets of multiple choice questions (MCQs) for testing reading comprehension have been created either for English or Swedish. However, for English, the use of synthetic MCQs has been explored for other domains, such as natural sciences (Le Berre et al., 2022) or factual QA (Puri et al., 2020).

QA for Swedish is an under-researched area with very few existing datasets. There has been an attempt to translate SQuAD (Rajpurkar et al., 2016), which does not contain MCQs, into Swedish[3] with no information on whether translations were manually checked. To the best of our knowledge, the only existing MCQ dataset in Swedish is SweQUAD-MC (Kalpakchi and Boye, 2021), which has been manually constructed.

## 3 Data collection

### 3.1 Textual materials

We have collected 96 texts of varying length, type and genre from the national tests of Swedish for Immigrants courses (swe. *SFI nationella prov*) using OCR. These texts have been specifically adapted to test reading comprehension of adult language learners of Swedish. The sought-after synthetic data should consist of MCQs for each given text, where each MCQ must fulfill a number of requirements:

1. there must be 4 alternatives;

2. only one alternative must be correct;

3. the other 3 alternatives must be wrong, but plausible;

4. the question must be answerable using the information in the given text.

For each text, a batch of MCQs fulfilling the requirements above should be generated and the number of MCQs in the batch should vary depending on the length of text: the longer the text, the more MCQs should be available. Additionally, the difficulty of MCQs in each batch should vary.

---

[3] https://github.com/Vottivott/swedsquad

## 3.2 GPT-3 hyperparameters

We have employed OpenAI's GPT-3 (Brown et al., 2020), more specifically version *text-davinci-003*, to generate synthetic data that fuifils the requirements from the previous section.

### 3.2.1 Prompt

Our approach to creating the prompt was to spell out the aforementioned requirements as clearly as possible. The results was the following prompt, which has been fed to GPT-3 **in Swedish**.

> Skriv $N_q$ olika läsförståelsefrågor med 4 alternativ (a, b, c, och d) och ge varje fråga en unik nummer (1, 2, 3, osv). Första alternativet (a) ska alltid vara rätt, medan de andra alternativen (b, c, och d) ska vara felaktiga, men troliga. Alla frågor måste kunna besvaras av den följande texten. Ordna frågor från den lättaste till den svåraste.

The number $N_q$ was selected based on the length of each text (the longer the text, the more MCQs we asked for) using the heuristic detailed in Appendix A.

To help non-Swedish-speaking readers, we also provide an English translation of the prompt below, but we emphasize again, that all input to GPT-3, *including* the prompt, was *in Swedish*.

> Write $N_q$ different reading comprehension questions with 4 alternatives (a, b, c, and d) and give each question a unique number (1, 2, 3, and so on). The first alternative (a) should be always correct, while the other alternatives (b, c, and d) should be wrong, but plausible. All questions must be answerable of the following text. Order questions from the easiest to the hardest.

We did **NOT** perform any extensive experimentation with prompt formulation. We have formulated the prompt in a way that it includes all aforementioned requirements in the most unambiguous way possible. Some parts of the requirements are ambiguous by necessity, for instance, the definitions of MCQ difficulty vary among researchers (see Section 4.1 for further discussion on the matter). The intention behind including the difficulty requirement into the prompt was to check whether GPT-3 could produce any variation at all when it comes to MCQ difficulty.

### 3.2.2 Generation hyperparameters

We did **NOT** perform any systematic search for the generation hyperparameters (e.g., temperature, top P for nucleus sampling, etc). Instead we used the default settings (listed in Appendix B), except for the extended maximum generation length to allow for longer texts and more MCQs.

The rationale behind this decision is that it is impossible to define the degree to which we want GPT-3 to generate repeated content. For instance, if the text consists of one and only sentence: "Stockholm is the capital of Sweden", then one of the few good reading-comprehension questions would be "What is the capital of Sweden?", with the correct answer "Stockholm". In this example, all words from the text are repeated in the question and the correct answer. One could, of course, paraphrase the question to some degree, but then that poses a risk of the question's meaning "drifting away". For instance, the question "What is the administrative center of Sweden?" is still a valid question, but it is neither equivalent to the original question, nor answerable from the given text.

## 4 Evaluation methodology

We are interested to know how well GPT-3 followed the instructions given in our prompt. For each given text, we investigated the following properties:

Q1. Were $N_q$ MCQs generated?

Q2. Did every MCQ include a stem and 4 alternatives?

Q3. Did the formatting conform to the requested one (MCQs are numbered, alternatives are labeled with letters a, b, c, d, etc)?

Q4. Were all MCQs distinct?

The next group of questions is interesting only for distinct MCQs with a stem and 4 alternatives. We will refer to these as *D-questions*, with "D" for "distinct".

D1. Were all stems grammatically correct and answerable after reading the text?

D2. For MCQs having stems compliant with the requirements in D1, were all alternatives grammatically correct and relevant?

The final 3 questions are interesting only for those cases where the answer was *yes* for both D1 and D2. We will refer to these as *R-questions*, with "R" for "relevant".

R1. Was only one alternative always correct, while the others were always wrong, but plausible?

R2. Was the correct alternative always a?

R3. Were the MCQs always ordered from the easiest to the hardest?

Although requiring some manual annotation, the questions above are all trivial to check, with the exception of R3, which is non-trivial since the concept of MCQ difficulty is not well-defined. In fact, MCQ difficulty depends on many things that are hard to keep constant, e.g., the reader's skills and background knowledge, whether the test is taken under time pressure, etc. For the purpose of this case study, we have relied on a definition of difficulty outlined in the section below and further detailed in Appendices C.1, C.2, and C.3.

### 4.1 MCQ difficulty

For defining MCQ difficulty we take inspiration from the methodology proposed by I. Kirsch and P. Mosenthal, which served as one the bases for the TOEFL 2000 (Jamieson et al., 2000) and PISA 2018 (OECD, 2019) reading literacy frameworks. In particular we consulted Kirsch and Mosenthal (1995), because this work specifically deals with assessing difficulty of multiple-choice questions.

Kirsch and Mosenthal (1995) have used the percentage $p_c$ of students who answered the question correctly[4] as a proxy for the MCQ difficulty. In an attempt to explain performance differences, they have defined a number of readability and reading process variables, and ran a regression using these variables as predictors of $p_c$. They found the following three variables to be particularly strong predictors (later referred to as *core predictors*):

- Type of Information (TOI)

- Type of Match (TOM)

- Plausibility of Distractors (POD)

---

[4]In Kirsch and Mosenthal (1995), this quantity is called *p-value*, but should not be confused with *p*-values from statistical hypothesis testing, which are also reported using *p*-notation

Inspired by Kirsch and Mosenthal (1995), we evaluated each of the core predictors on a scale from 1 to 5 using the following scoring rules:

- **TOI**: The more abstract the stem, the higher the score. Stems inquiring about concrete things like places or people will get a score of 1, whereas those asking about more abstract concepts will get increasingly higher scores, up to 5 for the most abstract concepts, like themes or patterns.

- **TOM**: The more inference required to match the information in the stem and the key to the text, the higher the score. This means a score of 1 for MCQs requiring simple string matching, up to a score of 5 for those matches requiring reading between the lines.

- **POD**: The closer distractors are to the key in the text, the higher the score. This means a score of 1 for MCQs with no distractors present in the text, up to a score of 5 in the cases where two or more distractors are close to the key in the text.

More precise definitions for scoring the core predictors are provided in Appendix C.1 for Type of Information, Appendix C.2 for Type of Match, and Appendix C.3 for Plausibility of Distractors.

# 5 Results

Recall that we collected 96 texts and asked GPT-3 to generate $N_q$ MCQs for each of them, where $N_q$ is calculated based on the length of each text (the longer the text, the more MCQs we asked for). In total, GPT-3 made 718 generation attempts. To answer all questions posed in the previous section, we have made all required manual annotations ourselves using an iterative annotation process (annotating – discussing issues – re-annotating). All annotations for this section have been performed using the Textinator[5] annotation tool (Kalpakchi and Boye, 2022).

Q1. Were $N_q$ MCQs always generated?
**Answer**: No, but very often (for 89.6% of the texts)

For 86 out of 96 texts, GPT-3 generated exactly the requested $N_q$ MCQs. The mismatch between

---

[5]To facilitate reproducibility, the exact details of the Textinator setup are available in the GitHub repository associated with this paper.

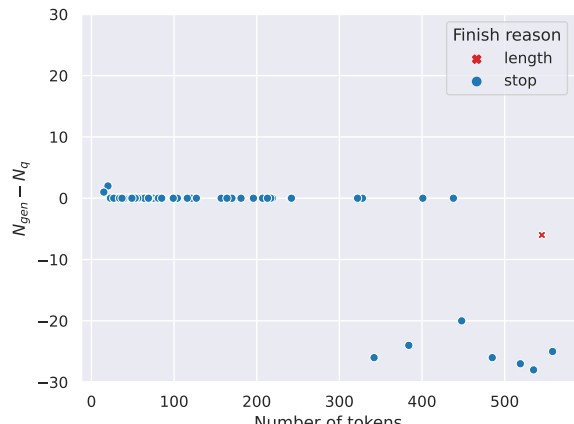

Figure 1: Scatterplot of the relation between the number of tokens (as provided by NLTK) and the size of MCQ number mismatch, $N_{gen} - N_q$

the number of generated MCQs $N_{gen}$ and $N_q$ is shown in Figure 1.

As can be seen, most of the mismatch happens for longer texts and there are mostly fewer MCQs generated than requested. One possible explanation could have been that GPT-3 simply did not have enough tokens in its context window. However, Figure 1 illustrates that in the vast majority of cases, GPT-3 stopped generating MCQs after reaching the stop token. In fact, only in one case was the generation interrupted because the context window was too short (GPT-3 failed even to produce a stem for this example). This means that **717 out of 718** generation attempts resulted in an MCQ.

Q2. Did every MCQ include a stem and 4 alternatives?
**Answer**: Yes

The only MCQ that did not was the one with the id 0_28, which was the only failed generation attempt discussed above. All other **717 MCQs** contained a stem with 4 alternatives.

Q3. Did the formatting conform to the requested one (MCQs are numbered, alternatives are labeled with letters a, b, c, d, etc)?
**Answer**: Yes, with some minor variations.

The stems were always numbered using Arabic numbers followed by a full stop. The alternatives were always formatted in the same way both within each MCQ and between all MCQs for each text. The formatting itself has slightly differed between the texts, using either small or capital letters

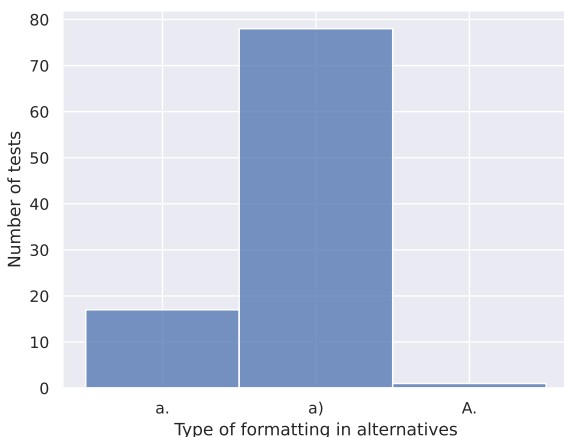

Figure 2: Distribution of the formatting types for the alternatives in each test (using the first alternative `a` as an example). All MCQs within the test were formatted in the same way.

from `a` to `d`, followed by either a right bracket or a full stop. The distribution of different formatting options is illustrated in Figure 2.

Q4. Were all MCQs distinct?
**Answer**: Mostly yes (around 4% duplicates).

MCQs can be duplicated to varying extents. We define the following cases, which we call *duplication levels*:

- **absolute** – when both the stem and all alternatives are the same (ignoring the punctuation), **and** the alternatives have been generated in the same order;

- **partial** – when either only the stem is the same, **or** both the stem and all alternatives are the same, but the alternatives have been generated in a different order;

- **paraphrased** – when the stem (and possibly a subset of alternatives) is a paraphrased version of the stem (and possibly a subset of alternatives) of the other MCQ(s).

If one MCQ is a duplicate of more than one MCQ, we take only the strongest duplication level into account. For instance, if X and Y are paraphrased duplicates, whereas X and Z are absolute duplicates, we include X as the case of absolute duplicates in the descriptive statistics.

**31 (4.32%) MCQs** turned out to be duplicates with a distribution of duplication levels provided in Figure 3. **As previously mentioned, all duplicates are excluded from further analysis.**

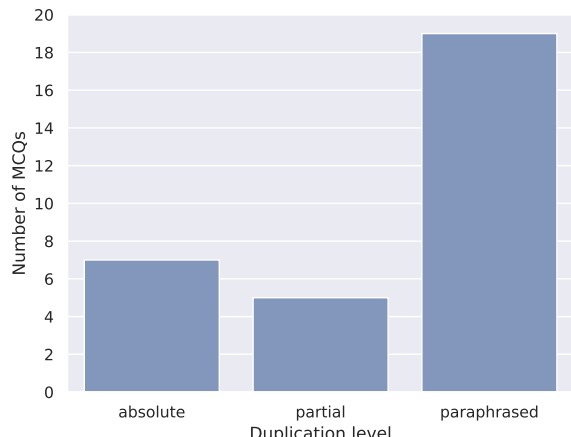

Figure 3: Distribution of duplicated MCQs (4.32% of all MCQs) per duplication level

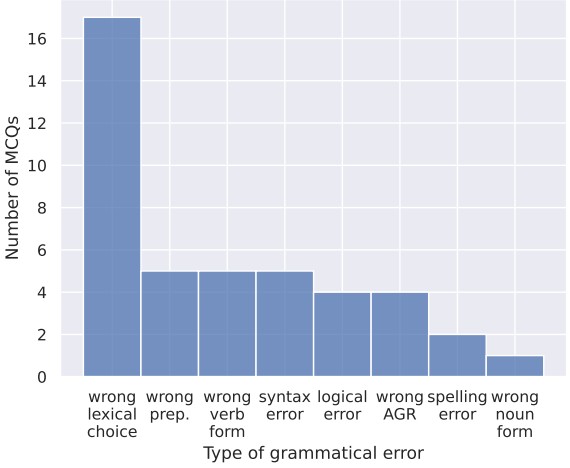

Figure 4: Distribution of grammatical error types for stems. "AGR" stands for "agreement", and "prep." – for "preposition"

D1. Were all stems grammatically correct and answerable after reading the text?
**Answer**: No (roughly 1 in 5 MCQs did *not* conform to these requirements)

There are multiple kinds of problems related to D1. The first problematic category includes ungrammatical stems, which we have classified further into the types of grammatical errors, shown in Figure 4. In total, **43 (6%) MCQs** had ungrammatical stems with a more detailed description and examples for each grammatical error type given in Appendix D.

The second problematic category concerns the stems that are grammatically correct, but *unanswerable* for the given text. For the purpose of the synthetic data at hand, we have defined the follow-

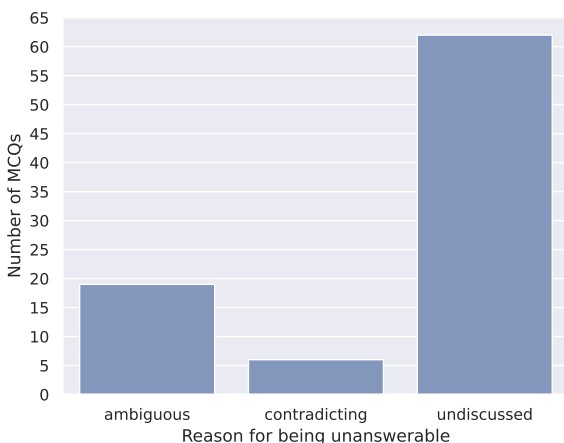

Figure 5: Distribution of reasons for being unanswerable.

ing reasons to classify a stem as unanswerable.

- **Contradictive**, meaning that a presupposition in the stem contradicts what is written in the text. For instance, suppose that in the text it is written "John was very happy to finally resign", and the stem is "Why was John sad about resigning?". Here the presupposition that John was sad is inconsistent with the text. Another example could include the text "John likes playing basketball, but his biggest hobby is tennis" and the stem "What is John's hobby?". This formulation of the stem presupposes that John has one hobby, which is not true and hence inconsistent with the text.

- **Undiscussed**, meaning that the text neither provides the information necessary to find the key for the stem, nor provides the way to reject all but one alternative, while providing some support for the remaining one. In either case the information in the stem does **NOT** contradict the text.

- **Ambiguous**, meaning that the information provided in the stem is not enough to choose one definite answer among the provided alternatives, i.e., different alternative(s) could be viewed as the key, depending on the interpretation of the stem.

**87 (12.13%)** MCQs were deemed to have unanswerable stems with a more fine-grained distribution depicted in Figure 5.

The last, but not least problematic category in D1 is that of grammatically correct stems that could be answered without reading the text. This category includes **20 (2.79%) MCQs**.

D2. Were all alternatives grammatically correct and relevant for the given stem and text?
**Answer**: No, but more than for stems (around 3 in 20 MCQs did *not* conform to the requirements above).

Similarly to D1, there are multiple kinds of problems related to D2. One problem is that of ungrammatical alternatives, which uses exactly the same categorization as for D1 (detailed and exemplified in Appendix D) with one additional category: "tautology". In total **10 (1.39%) MCQs** with grammatically correct stems had at least one ungrammatical alternative, with the error type distribution provided in Figure 6.

The other problem concerns cases when the alternatives are grammatically correct, but irrelevant for the given text. For the synthetic data at hand, we have defined the following reasons to judge the alternatives as irrelevant for the given text.

- **Misfocused**, meaning that at least one of the alternatives does not provide the type of information, requested in the stem. One example of such inconsistency would be the stem "What is the capital of Sweden?", accompanied by the alternative "John Lennon". Note that even if the correct answer, "Stockholm", is within the provided 4 alternatives, but so is "John Lennon", the MCQ will still be categorized as misfocused. The rationale is that in

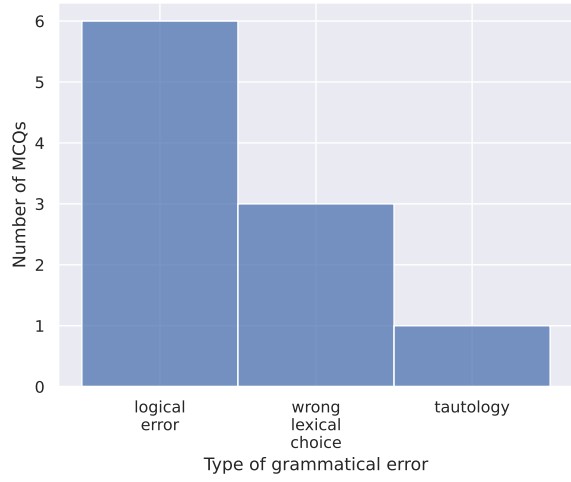

Figure 6: Distribution of grammatical error types for alternatives.

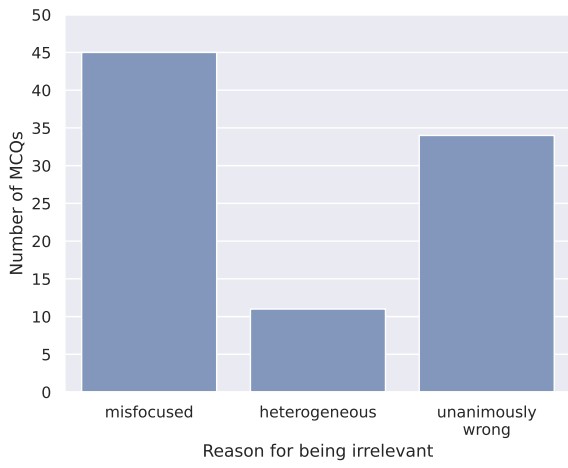

Figure 7: Distribution of reasons for being irrelevant.

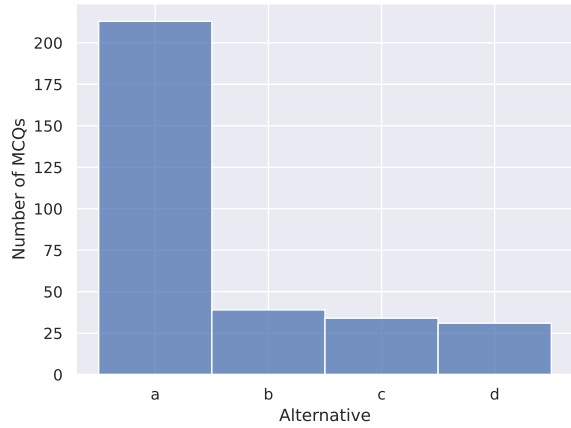

Figure 8: Distribution of the positions of correct alternatives

such cases, the effective number of alternatives becomes less than 4 and thus it becomes easier to guess the correct answer.

- **Heterogeneous**, meaning that one or more of the provided alternatives stick out and thus provide a potential clue for the students. One example would be the stem "Where is Nobel Museum located?" and the alternatives "Stortorget 2, 103 16 Stockholm", "Gothenburg", "Uppsala", "Copenhagen". The first alternative is clearly different from the others and is also the correct answer in this case.

- **Unanimously wrong**, meaning that neither of the provided 4 alternatives can be considered correct (the key).

**90 (12.55%) MCQs** were judged to be irrelevant with the distribution of reasons for irrelevancy depicted in Figure 7.

To summarize, the R-questions will be evaluated only on the MCQs that didn't have any problems so far. This includes $717 - 31 - 43 - 87 - 20 - 10 - 90 = 436$ MCQs (60.81%).

R1. Was only one alternative always correct, while the others were always wrong?
**Answer**: No, around 3 in 10 of the remaining MCQs (or 3 in 20 in total) had problems.

**119 (16.6%)** of the remaining 436 MCQs had more than one correct answer, which leaves us with 317 MCQs (44.21%) to be tested for the remaining conditions.

R2. Was the correct alternative always `a`?
**Answer**: No, a bit more than 3 in 10 of the remaining MCQs (or 3 in 20 in total) had `b`, `c`, or `d` as the correct alternative.

The distribution of positions of correct alternatives for the 317 MCQs remaining after R1 is provided in Figure 8. For **213 MCQs (29.71%)** the alternative `a` was correct, whereas all the other were wrong.

R3. Were the MCQs always ordered from the easiest to the hardest?
**Answer**: No, but for 27 texts they were!

For this part of the analysis, we have included all 317 MCQs with exactly one correct answer (no matter `a` or not) and without any problems spotted before R2. Notably, 6 texts have lost **all** their MCQs, so these 317 MCQs are spread over 90 out of the initial 96 texts.

We have then annotated each MCQ using the MCQ difficulty scheme outlined in Section 4.1 (and detailed in Appendices C.1, C.2, and C.3). The distribution of total MCQ difficulty is shown in Figure 9. Recall that the minimum possible MCQ difficulty is 3 points, whereas the maximum is 15 points. Each column in Figure 9 represents one of the 90 texts and each row is an MCQ generated for this texts. The MCQs are ordered in the order of generation from bottom to top (so the first row from the bottom indicates the first MCQ generated by GPT-3). Grey cells indicate MCQs excluded prior to R2.

If GPT-3 followed the prompt and ordered MCQs from easiest to hardest, one would expect

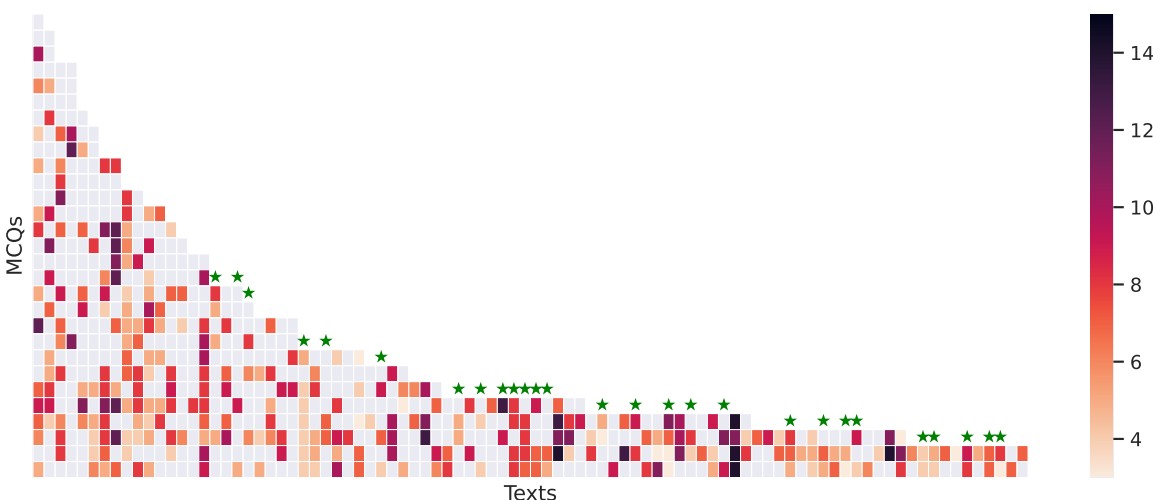

Figure 9: Heatmap of MCQ difficulty. Each column represents one of the 90 survived texts. Each cell in a column represents an MCQ generated by GPT-3 for this text. Grey cells indicate the MCQs excluded because of insufficient quality, whereas cells of other colors represent the accepted MCQs. Difficulty ranges from 3 to 15 and is represented by colors according to the legend on the right. Texts are ordered by their length (from the longest to shortest), which is directly proportional to the requested number of MCQs. MCQs are ordered by their generation order (those generated first reside in the first row from the bottom). The columns with green stars indicate the texts which had more than 1 accepted MCQ generated by GPT-3 in the order of their difficulty (from easiest to hardest).

the whole heatmap (except the grey cells) to follow the same coloring as the legend. The easier MCQs with difficulties close to the theoretically minimal 3 points should be at the bottom of the chart in light colors. The hardest MCQs should be on top of every column in dark colors (with difficulties close to theoretically maximal 15 points). However, Figure 9 shows neither this pattern, nor any pattern at all. Nevertheless, if we consider texts which have more than one survived MCQ (**72 out 90 texts**), then MCQs were ordered in the non-decreasing order of difficulty for **27 texts** (marked with green stars in Figure 9).

## 6 Discussion and conclusions

GPT-3 has been able to generate around 30% of MCQs that conformed to all criteria (excluding ordering by difficulty), and 44% of MCQs which were of sufficient quality (also excluding the requirement that `a` is the correct answer). Together with our additional annotations, detailed with examples in Appendix E, these 44% of MCQs constitute *Quasi*, the first synthetic dataset of MCQs for testing reading comprehension of adult language learners of Swedish (available at `https://github.com/dkalpakchi/Quasi/blob/main/annotated/quasi.json`).

The fact that 44% of MCQs turned out to be of sufficient quality is impressive, given that (a) it is zero-shot, and (b) only 0.11% of GPT-3's training data was in Swedish. Although GPT-3 did not manage to order MCQs from easiest to hardest for most of the texts, the model could still generate MCQs of varying difficulty levels. The easiest MCQ scored theoretically minimal 3 points, whereas the hardest scored 14 (just 1 off from the theoretical maximum!).

That said, as 56% of MCQs turned out to be of insufficient quality (available at `https://github.com/dkalpakchi/Quasi/blob/main/annotated/poor_quality.json`), sometimes for subtle reasons, manual curation is not only desired, but is in fact *required* (not least to identify the correct alternative).

**Why not ask GPT-3 to choose the correct alternative?** One counter-argument is that it would consume more tokens, which leaves less tokens for MCQs, and leads to higher costs. Another reason is that there is no convincing argument why GPT-3 would be able to always provide the correct answer. If it could, then it should have been able to put it as alternative `a` all the time, which it did not. Furthermore, it should have been able to *always* generate only one correct answer, which it did not

do for 16.6% of MCQs. In fact, this finding is in line with the previously published evaluation of a BERT-based model for generating distractors in Swedish (Kalpakchi and Boye, 2021), where the most frequent reason for rejecting distractors was that they were not wrong (leading to more than 1 correct answer).

**Could GPT-3 handle OCR errors, if there were any?** Yes, it could! To give an example, one of the e-mail addresses in one of the texts was incorrectly recognized by the OCR system as *"ifhs.info Qimh.se"*, which we unfortunately didn't notice. GPT-3 was still able to generate *"ifhs.info@imh.se"* as one of the alternatives. This is most probably, because there was *"e-post:"* (eng. *"e-mail:"*) before this string, which the GPT-3's attention mechanism was able to capture. That said we didn't do any rigorous evaluation to quantify how well GPT-3 can mitigate OCR errors, so the caution is advised when trying to generalize from this insight.

## Acknowledgments

This work was supported by Digital Futures within the project SWE-QUEST. We would like to thank Mariia Zyrianova for helpful discussions on the intricacies of the Kirsch scheme. We would also like to thank the anonymous reviewers for their helpful comments and suggestions.

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

## A Heuristic for choosing the number of generated MCQs

First we have calculated the average length of the sentence in characters for different sources of text (forum, news, blogs, etc), which could be extracted by multiplying $\bar{W}$ by $\bar{C}$ from Table 1. Then for each text $T$ (belonging to category $cat_T$) we have calculated $N_q^T$ as follows:

$$N_q^T = \alpha \frac{C_T}{\sum_{c \in cat_T} \bar{W}_c \cdot \bar{C}_c}, \qquad (1)$$

where $C_T$ is the number of characters in $T$, $\bar{W}_c$ ($\bar{C}_c$) is the average number of words (characters) per sentence in the corpus $c$ belonging to the category $cat_T$ (if a text did not belong to any category, we set $cat_T$ to all categories), $\alpha$ is the assumed number of MCQs to be asked per sentence.

Choosing $\alpha$ is a bit tricky. In reality one can ask way more than 1 MCQ per sentence, but then not all sentences are worth asking even 1 MCQ. In hopes that these two groups cancel each other out, we have assumed $\alpha = 1$, meaning 1 MCQ per sentence, for the purpose of this article.

## B Hyperparameter settings for GPT-3

We have used OpenAI's *text-davinci-003* model with the following generation hyper-parameters:

- temperature of 0.7

- "top p" (for nucleus sampling) of 1

- frequency and presence penalties of 0

- "best of" being equal to 1

- no custom stop sequences

- maximum length of 2048

## C Scoring rules for process variables included in MCQ difficulty calculation

Recall that the core predictors found by Kirsch and Mosenthal (1995) are Type of Information (TOI), Type of Match (TOM), and Plausibility of Distractors (POD). The way these predictors were proposed to be operationalized is different depending on the nature of the provided textual material. More specifically, Kirsch and Mosenthal (1995) distinguished between the following two types of materials:

| Corpus $c$ | Source | $|W_c|$ | $\bar{W}_c$ | $\bar{C}_c$ |
|---|---|---|---|---|
| Familjeliv | forum | 885M | 12.56 | 4.51 |
| Flashback | forum | 711M | 12.92 | 4.67 |
| Bloggmix | blogs | 375M | 13.82 | 4.69 |
| Webbnyheter | news | 87M | 15.31 | 5.42 |
| SVT | news | 179M | 13.65 | 5.40 |
| Wiki | info | 314M | 10.67 | 5.63 |

Table 1: Corpus statistics for deciding the value of $N_q$. $|W|$ denotes a number of words in a corpus, $\bar{W}$ – the average number of words per sentence, $\bar{C}$ – the average number of characters per word

- **prose**, which refers to texts that consist of sentences grouped into paragraphs, in particular narrative and expository texts were considered;

- **documents**, meaning any kind of material where the structure of the document provides extra information for understanding the contents, for instance, e-mails (with address headers and footers), articles (with headlines), reports (with tables and graphs), advertisements, schedules, etc.

In the following section we discuss how we adapted the scheme proposed by Kirsch and Mosenthal (1995) to the needs of this article.

### C.1 Type of Information

Kirsch and Mosenthal (1995) defined *Type of Information (TOI)* as the nature of what the readers are asked to identify in each given stem. The basic rule is that the more concrete the TOI, the easier the MCQ is, whereas more abstract TOI lead to more complex MCQs. The scoring rules are as follows.

- person, animal, or thing, score 1;

- amount, time, attribute, action, location, score 2;

- manner, goal, purpose, condition, or predicate adjective, score 3.

- cause, result, reason, evidence, or theme, score 4.

- equivalent, score 5.

The same rules apply for both prose and document materials.

### C.2 Type of Match

Kirsch and Mosenthal (1995) defined *Type of Match (TOM)* in terms of processes used to relate information in the stem and the key to information in the textual material (prose or document). This is the most complex part of the scheme for evaluating MCQ difficulty, which we have simplified and took only the core aspect of it.

For the sake of brevity, we denoted the relation between a stem and the textual material as S-T, and the relation between a key and the textual material as K-T. For prose tasks, the majority of which were multiple-choice questions (MCQs),

| Prose | Document | Score |
|---|---|---|
| when `S-T` **and** `K-T` are both `LIT` or `SYN` | | 1 |
| when `S-T` **or** `K-T` requires `LLTI`, while the other requires `LIT` or `SYN` match | | 2 |
| when `S-T` **and** `K-T` both require `LLTI` | | 3 |
| when either `S-T` **or** `K-T` requires `HLTI` | | 4 |
| when `S-T` **and/or** `K-T` requires `HLTI`, but the nature of the corresponding relation(s) needs to be defined by the reader | when `S-T` **and/or** `K-T` requires inferring a causal pattern or trend, or making a unique inference based on prior knowledge | 5 |

Table 2: Scoring rules for Type of Match detailed for prose and document materials. `S-T` denotes the relationship between the stem and the text, and `K-T` between the key and the text, `LIT` stands for "literal", `SYN` – for "synonymous", `LLTI` – for "low-level text-based inference", whereas `HLTI` – for "high-level text-based inference".

we have adopted the scoring rules of Kirsch and Mosenthal (1995) as they are (see Table 2).

The scoring rules for document tasks were defined using many special-case rules. However, Kirsch and Mosenthal (1995) note that *many* of the document tasks did not use a multiple choice format, whereas in our case *all* tasks are guaranteed to be MCQs. Henceforth, instead of adopting, we chose to *adapt* by keeping as many applicable aspects of the rules for prose texts, as possible.

A clear similarity between prose and document scoring rules is that tasks requiring literal or synonymous match are still easier than those that need low-level text-based inference, which, in turn, are easier than those requiring a high-level text-based inference. Hence we decided *to keep the first 4 scoring rules as they are.*

One clearly different thing is the definition of the final level (when the MCQ should be awarded 5 points). We adopt this difference, but slightly adapt it, as shown in Table 2.

Unfortunately, Kirsch and Mosenthal (1995) do not provide clear definitions of what low-level or high-level inference mean, or where the border between synonymous match and low-level inference is. Hence, for the sake of this article, we have devised the following definitions based on examples of scoring MCQs, provided by Kirsch and Mosenthal (1995), and common sense.

**Literal match** (`LIT`) entails that the required information exists in the given textual material word-by-word. When applying this definition, it is allowed to ignore:

- question words/phrases, for instance, *"Vad"* (eng. *"What"*), *"Hur mycket"* (eng. *"How much"*), *"I vilket land"* (eng. *"In what country"*);

- articles, for instance *"en / ett"*, *"den / det / de"*, *"denna / detta / dessa"*;

- changes in the word form, when the word stem[6] remains the same (see examples in Table 3);

- changes between parts of speech (e.g., nominalization, adjectivization) when the word stem remains the same (e.g., *"drömmar"* and *"drömmer"*, *"samlas"* and *"samling"*)

For documents, there is a special case of `LIT` when matching information requires identifying structural part(s) in a document with a widely accepted structure. Such documents include, but are not limited to, e-mails, letters, blog posts, schedules. To exemplify, the question "Who is the sender of an e-mail?" requires the reader to locate the signature at the end of the email.

**Synonymous match** (`SYN`) encompasses cases when one word is substituted for another word with a similar meaning and the same grammatical features (part of speech, voice, inflection, number, etc). One or more such substitutions are allowed. Additionally, the following cases are included in this category, although they are not typically counted as synonyms in linguistics.

---
[6]not to be confused with the stem of an MCQ

| TOM | Description | Example |
|-----|-------------|---------|
| LIT | conjugations of regular verbs | swe. *vänta / väntar / väntade / väntat* 
 eng. *wait / waits / waited / waited* |
| LIT | conjugations of some irregular verbs to Present Simple | swe. *gå / går, stå / står, ge / ger* 
 eng. *go / go, stand / stand, give / give* |
| LIT | noun inflections | swe. *bil / bilen / bilar / bilarna / bils / bilens / bilars / bilarnas* 
 eng. *car / the car / cars / the cars / car's / the car's / cars' / the cars'* |
| LIT | adjective inflections | swe. *stor / stort / stora* 
 eng. *large / large / large* |
| SYN | conjugations of strong verbs | swe. *finna / finner / fann / funnit, bryta / bryter / bröt / brutit* 
 eng. *find / find / found / found, break / break / broke / broken* |
| SYN | most conjugations of irregular verbs | swe. *går / gick / gått, står / stod / stått, ger / gav / givit* 
 eng. *go / went / gone, stand / stood / stood, give / gave / given* |

Table 3: Examples of word form changes allowed different Type of Match levels

- Changes in the word form, when the word stem becomes different (see examples in Table 3).

- Using comparative and superlative adjectives, e.g., *"god / bättre / bäst"* (eng. *"good / better / best"*.

- When a word is matched to a part of a compound, e.g., *"kurserna"* in the stem and *"kursveckor"* in the text.

- Using abbreviations, e.g., *"tel."* for *"telefonnummer"*, *"kl."* for *"klockan"*, *"Feb"* for *"Februari"*.

- When numbers are written as words and vice versa, as well as colloquial names for numbers (*"a pair"* meaning 2).

**Low-level text-based inference** (LLTI) includes cases when:

- the required information needs to be "collected" from multiple sentences (e.g., co-reference resolution);

- requires local (within sentence) reasoning (e.g., if the text says *"John is older than Mary"*, while one of the alternatives is *"Mary is younger than John"*);

- a word is substituted for another word with a different word stem, but a similar meaning, but different grammatical features (part of speech, voice, inflection, number, etc);

- a word is substituted by a phrase or vice versa;

- compounds (swe. *sammansättningar*) that are split into separate words, for instance, exchanging *"grundutbildning"* with *"grundläggande utbildning"* (note that these are very rare in English, but quite common in Swedish);

- hierarchical relationships, e.g. *"basketball"* and *"sport"*;

- the format needs to be recognized, e.g. that *"name@example.com"* is an e-mail;

- a non-matching word denotes whether the information should be included in/excluded from a document (or a part of a document).

**High-level text-based inference** (HLTI) when it is required to link multiple paragraphs of text. These cases include, but are not limited to:

- counting entities (if they are not already counted in the text), such as in the stem *"How many countries are represented in the event"*;

- reading between the lines to find out the information, as in *"Why did John write this letter?"*;

- using specific prior knowledge about content or structure of the text, for instance, when one writes *"Otto, 27"* at the end of the post on social media means that *"27"* is most probably his age, or when it's written *"Opening*

*hours 11 - 21"*, it means that the closing time is 21:00;

- asking whether the information is included in the text, e.g. *"What is not a hobby of John?"*.

The last level, for the score of 5, requires the reader to define the nature of the `S-T` and/or `K-T` relations. In particular, this included the cases when the reader needs to

- provide an interpretation of a phrase based on the information in the text, e.g., *"Vem av personerna i texten är mest förespråkare för förbud?"* (eng. *"Which of the people in the text advocates the most for the ban?"*);

- recognize the stance of a person in the text, e.g. *"Hur resonerar Joel Marklund kring ekologiska produkter?"* (eng. *"How does Joel Marklund reason about the ecological products?"*).

For all TOM levels, pleonasms (meaning redundant linguistic expressions that are unnecessary to comprehend the stem) should be *ignored*. For instance, consider the key is *"make demands on the children but show them love"*. The pronoun *"them"* is only there because of grammar, otherwise it is apparent from the context that love should be directed towards children, even without the pronoun. Note that in MCQs expressions might become pleonastic with respect to the given alternatives. For instance, consider the sentence *"Welcome to the interview on Wednesday 6/2 at 15:00"* and the following MCQ.

> What is the date for the interview?
>
> a) Thursday 7/2
>
> b) Wednesday 6/2
>
> c) Wednesday 15/2
>
> d) Thursday 6/2

Obviously, the word *"date"* is not mentioned in the text and one needs to know what the date is, so this appears to be a case of `HLTI`. However, given the alternatives, one doesn't need to understand the word *"date"* and suddenly the match between the stem and the text gets downgraded to `LIT`.

To give another example consider the following e-mail.

> Hi,
>
> I was forced to pay $20 extra for the delivery of the laptop, which I think is unacceptable!
>
> Best regards, Martin Jones

If we analyze the MCQ below at a first glance, *"Martin Jones"* is not mentioned in the sentence about extra $20 payment. Instead *"I"* there should be resolved to *"Martin Jones"*, so it seems like a case of `LLTI`. However, Martin Jones is the only person that is in fact mentioned in the text, so mention of his name in the stem becomes pleonastic and hence the MCQ again gets downgraded to `LIT`.

> How much was Martin Jones forced to pay?
>
> a) $20
>
> b) $15
>
> c) $40
>
> d) $2

What these two examples show is that the judgement of Type of Match level if *extremely* text-dependent and one and the same MCQ could get different TOM-score, depending on the text at hand.

## C.3 Plausibility of Distractors

Inspired by (Kirsch and Mosenthal, 1995), we make use of an implicit tree structure for both prose and document materials. Each node of such a tree should contain a unit of information that cannot be split further into independent units. The only type of nodes in prose texts are paragraphs, whereas nodes in documents are generalized paragraphs by nature, but could also contain more structured and/or graphical material (such as charts, tables, maps, lists, etc).

Given the surface form of the correct answer, we define *the answer node* (`AN`) as the first node in the BFS traversal of the tree corresponding to the textual material, containing information supporting the correct answer.

Since *many* of the document tasks did not use a multiple choice format, as noted by Kirsch and Mosenthal (1995), the rules for scoring POD for document materials must be adapted. If the format is not MCQs, then it is only relevant to look into *distracting information* in the text, i.e., pieces of

| Prose | Document | Score |
|---|---|---|
| | There is no distracting information in the text | 1 |
| | `DIS` are `LIT` **or** `SYN` match to the information **not** in `AN` | 2 |
| | `DIS` represent `PII` **not** based on information related to `AN` | 3 |
| | One `DIS` contains information that is related to the information in `AN` | 4 |
| | Two or more `DIS` contain information that is related to the information in `AN` | 5 |
| | One or more `DIS` represent `PII` based on information outside of the text | 5 |

Table 4: Scoring rules for Plausibility of Distractors detailed for prose and document materials. `DIS` stands for "distractor(s)", `LIT` – for "literal", `SYN` – for "synonymous", `PII` – for "plausible invitied inferences", and `AN` – for "answer node".

text that provide plausible grounds, although they are still not correct. In stark contrast, MCQs already provide a number of alternatives, which the reader is forced to choose between. Hence the distracting information is only relevant if one of the distractors in the alternatives relies on it. Keeping that in mind, we have adapted the POD scoring rules for prose texts to the document texts by generalizing from paragraphs to nodes (see Table 4).

## D   Grammatical error types

The following is a list of grammatical error types, which we adopted for this article. Note that this is *not* an exhaustive list of grammatical error types, but very much specific to the synthetic data at hand.

- wrong verb forms, such as *"meddelar"* in the stem *"Vilka problem kan man meddelar om man har ett akut problem?"*;

- wrong noun forms, such as wrong case;

- wrong prepositions, such as *"hos anläggningen"* instead of *"i anläggningen"*;

- wrong grammatical agreement (AGR), such as *"en krav"* instead of *"ett krav"*, or *"det minst antalet"* instead of *"det minsta antalet"*;

- syntax errors, most often errors in constructions of sentences, e.g., *"Är kursbok och arbetsmaterial ingår i kursavgiften?"* (eng. *"Are the course book and work material includes in the course fee?"*);

- spelling errors, such as *"addressedes"* instead of *"addresserades"*, or *"städt"* instead of *"städat"*;

- wrong lexical choice, when a word should not be used in the provided context, for instance the stem *"Vilka huvudroller är med i Lyckliga dagar?"* (eng. *"Which main roles participate in The happy days"*), or using the pronoune *"deras"* instead of *"sina"*;

- logical errors, when a word/phrase is used in a way that does not conform to its properties, for instance *"cykelbana"* in the stem *"I vilken sorts transportmedel finns en cykelbana?"* (eng. *"In what kind of transport does the bicycle lane exist?"*), or *"lokalen"* in the alternative *"lokalen tar för lång tid att spela"* (eng. *"the premises take too long to play"*);

- tautology, such as *"poetiska dikter"* (eng. *"poetic poems"*).

## E   Additional annotations for Quasi

In addition to the annotations necessary for evaluating the difficulty of each MCQ in Quasi, we also provide the following annotations (exemplified in Figure 10):

- the phrase/sentence that serves as the basis for the key;

- the phrase/sentence that serves as the basis for each distractor (for those distractors that actually use information from the text);

- the answer nodes that the learner must read in order to answer the question.

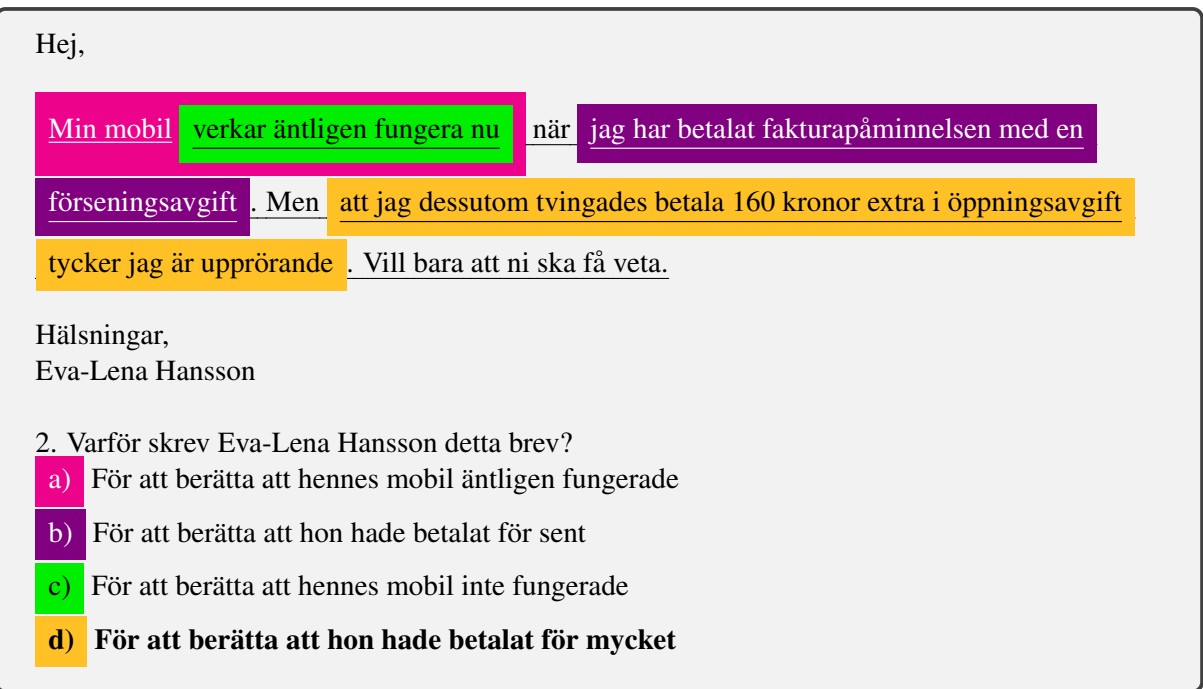

Figure 10: Example of additional annotations in Quasi. The bases for each alternative are highlighted in the text with the corresponding color. The required answer nodes are underlined. The key (correct alternative) is highlighted in bold.