# OpenReview forum: "Quasi: a synthetic Question-Answering dataset in Swedish using GPT-3 and zero-shot learning"
_NoDaLiDa/2023/Conference — NoDaLiDa 2023_

### Official Review · Reviewer_AJiM · 2023-02-24
**This paper describes an approach to produce Swedish multiple choice questions for reading comprehension texts using GPT-3**

**Rating:** 7
**Confidence:** 4

**Review:**

The paper describes a method for using GPT-3 to produce multiple choice questions (MCQ), such that only one answer out of four is correct and the other alternatives are so-called distractors. The MCQs can be used to assess reading comprehension skills among Swedish L2 learners. The evaluation shows that GPT-3 does a decent job: 44% of the generated MCQs are of "sufficient" quality (30% being almost fully in accordance with the criteria).

I think the paper is generally well written, with clear explanations and a logical flow of thought. Including the appendices, the paper is 14 pages long. The authors could consider whether the main text could be shortened to some extent and some crucial aspects presented in the appendices could be incorporated in the main text. There are a large number of examples, which is good in principle, but the type of examples I was most interested in finding, that is, actual MCQs produced by the model were scarce or non-existent. Maybe more space could be used for those rather than describing all possible error types?

The authors could also mention the existence of the specific GPT-3 for the Swedish language: GPT-SW3. How would that one have performed in comparison to the general, multilingual GPT-3 that is used in the paper? The related works section is very short. Even if corresponding work does not exist even for English(?), are there some similar approaches and what were the results there?

**Paper Type:**

Long paper

---

### Official Review · Reviewer_S2Ts · 2023-03-10
**Review of Quasi: A synthetic Question-Answering Dataset in Swedish using GPT-3 and Zero-Shot Learning**

**Rating:** 4
**Confidence:** 5

**Review:**

# Summary

The paper uses GPT-3 to generate a synthetic dataset of multiple-choice questions for text from the national tests from the ‘Swedish for Immigrants’ course. The authors prompt the model for a set of multiple-choice questions for each text, where each question should have four answer choices. Of the four possible answers, one should be correct, and the rest should be plausible but incorrect. All questions should be answerable with the given text. A heuristic based on text length determines the number of questions generated per text. The authors use a thorough evaluation methodology to determine the quality of the generated questions. The evaluation shows that 44% of the generated questions are of sufficient quality. Still, the model struggles with generating questions according to the prompted format. In particular, it generates questions with multiple correct answer choices and unanswerable questions. The authors find the results impressive, as the model is not specifically trained in Swedish, and the generations are done under a zero-shot setting. However, they caution that human curation of the data is still necessary.

# Strengths

A clear strength of the paper is the well-designed methodology used to determine the quality of the generated multiple-choice questions. The authors establish clear criteria for evaluating the generated questions, such as the presence of four alternatives and the plausibility of the incorrect answers. Additionally, the authors provide detailed explanations of the criteria in the appendix, making it easy for the reader to understand the evaluation process.

Another strength of the paper is the thorough analysis of the results. The authors carefully examine the limitations and failures of the generative model and present their findings in a clear and organized manner. The authors show how the model struggles to generate questions that meet the criteria outlined in the prompt and provide concrete examples of how the model fails to meet the requirements. This analysis provides valuable insights into the limitations of generative models for synthetic dataset generation, which can guide future research in the area.

Overall, the use of a well-designed methodology and thorough analysis of the results contributes to a well-rounded and informative paper that provides valuable insights for future work in the field of generative models and synthetic dataset generation – a field that is becoming highly relevant as generative models are becoming more capable in zero- and few-shot settings.

# Weaknesses

One major weakness of this paper is the absence of a dedicated section for conclusions and suggestions for future work. Although the authors thoroughly analyse their results, they fail to summarise their findings clearly and concisely. A separate section for conclusions and suggestions would have helped to tie the paper together and provide the reader with a better understanding of the paper's implications.

One of the paper’s aims is to create a synthetic dataset for ‘testing reading comprehension of adult language learners of Swedish’, but the authors do not adequately address whether the generated dataset is actually useful for that. The authors should at least have addressed this question in their conclusion and provided some empirical evidence to support the usefulness of the dataset for this specific application.

Additionally, the paper makes several unsupported claims that require further experimentation or citation to be convincing. For example, the claim that only 0.11% of the training data for GPT-3 is Swedish is not supported by any reference or citation. The same is true for the statement that cross-lingual learning has likely happened during training. The authors also suggest that the model's inability to always put the correct answer as the first answer choice implies that it would not be able to answer the question correctly. While this makes intuitive sense, it might not be true in practice due to the prompt sensitiveness of these models. It is a hypothesis that requires further verification. The authors also claim that manual dataset curation is required, but there are no experimental results to back that up. Also, automatic filtering methods could be explored (and have been in other works on synthetic dataset creation).

Overall, while the paper makes valuable contributions to synthetic dataset generation, it would benefit from a more thorough and critical examination of its claims and conclusions.

The related work section is thin and could benefit from including more information on synthetic dataset generation and MCQ tasks generally. Works that could be relevant to cite are:

* [Symbolic Knowledge Distillation]( https://aclanthology.org/2022.naacl-main.341.pdf)
* [Training Question Answering Models from Synthetic Data]( https://arxiv.org/pdf/2002.09599.pdf)
* [Unsupervised Multiple-Choice Question Generation for Out-of-Domain Q&A Fine-Tuning](https://aclanthology.org/2022.acl-short.83.pdf)

# Feedback on writing:

 The paper contains several typos. Here are a few:

* Section 2, last paragraph: There → The
* Section 3.2, first paragraph: fuifills → fulfils
* Section 3.2.1, first word: Out → our

The paper could be more concise and consistent in presenting information, such as using a single format when presenting percentages and ratios. The results are presented with percentages, counts, and ratios. Using one or two together might sound repetitive, but it makes it easier for the reader not to switch between different modes of parsing how many of the samples were correct.


**Paper Type:**

Long paper

---

### Official Review · Reviewer_9AEp · 2023-03-10
**A GPT-3 created dataset for Swedish QA**

**Rating:** 9
**Confidence:** 3

**Review:**

In "Quasi: a synthetic Question-Answering dataset in Swedish using GPT-3 and zero-shot learning" the authors introduce a new dataset for Swedish QA that has been created with the help of GPT-3.

Even though 0.11% of GPT-3's training data was in Swedish, the model is capable of handling Swedish texts.
The authors feed GPT-3 with texts from a Swedish language test, and task the model to create a question with multiple reasonable answers, of which only one can be correct.
The author's goal with this paper is not only to "lazily" create a new dataset, but even more to evaluate the capability of GPT-3 to create this dataset.
For this the authors do a thorough error analysis answering multiple clearly laid out questions, that are a good guideline for others trying to use GPT-3 or other models for similar work.

## Pros
- innovative use of GPT-3
- new dataset
- invites for imitation and gives good guidance
- helpful appendix

## Cons
- no discussion of the potential problems of this approach
- difficulty ordering seems a bit arbitrary as a requirement, but is somewhat interesting to see how much GPT-3 still manages to follow



**Paper Type:**

Long paper

---

### Decision · Program_Chairs · 2023-03-17

Accept